# Ambient Temperature and the Frequency of Subsequent Heart Failure Decompensations in an Emergency Department

**DOI:** 10.3390/biomedicines13051054

**Published:** 2025-04-27

**Authors:** Hermann Stefan Riepl, Viktoria Santner, Nora Schwegel, Viktoria Hoeller, Markus Wallner, Ewald Kolesnik, Dirk von Lewinski, Klemens Ablasser, Philipp Kreuzer, Klaus Zorn-Pauly, Faisal Aziz, Harald Sourij, Andreas Zirlik, Dieter Platzer, Nicolas Verheyen

**Affiliations:** 1Division of Cardiology, University Heart Center and Department of Internal Medicine, Medical University of Graz, 8010 Graz, Austria; stefan.riepl@medunigraz.at (H.S.R.); viktoria.santner@medunigraz.at (V.S.); nora.schwegel@medunigraz.at (N.S.); viktoria.hoeller@medunigraz.at (V.H.); markus.wallner@medunigraz.at (M.W.); ewald.kolesnik@medunigraz.at (E.K.); dirk.von-lewinski@medunigraz.at (D.v.L.); klemens.ablasser@medunigraz.at (K.A.); andreas.zirlik@medunigraz.at (A.Z.); 2Emergency Medicine Unit, Department of Internal Medicine, Medical University of Graz, 8010 Graz, Austria; philipp.kreuzer@uniklinikum.kages.at; 3Gottfried Schatz Research Center for Cell Signaling, Metabolism and Aging, Division of Medical Physics and Biophysics, Medical University of Graz, 8010 Graz, Austria; klaus.zornpauly@medunigraz.at (K.Z.-P.); dieter.platzer@medunigraz.at (D.P.); 4Trials Unit for Interdisciplinary Metabolic Medicine, Division of Endocrinology and Diabetology, Department of Internal Medicine, Medical University of Graz, 8036 Graz, Austria; faisal.aziz@medunigraz.at (F.A.); ha.sourij@medunigraz.at (H.S.)

**Keywords:** heart failure decompensation, climate change, temperature, heart failure subtype, continental climate

## Abstract

**Background/Objectives:** The impact of cold temperature on heart failure (HF) decompensations in continental climate zones is unclear. We aimed to evaluate the association between daily temperature and the subsequent frequency of HF decompensations in an emergency department (ED) in Eastern Austria. **Methods:** A systematic retrospective medical chart review of all admissions to the ED of a tertiary care center within 12 months was conducted. Maximal daily temperature and further meteorological data were obtained from the National Institute for Meteorology and Geodynamics. **Results:** Among 32.028 ED admissions, there were 1.248 HF decompensations. Median maximal daily temperature ranged from 4.3 °C in January to 28.7 °C in August, and the frequency of decompensations ranged from 65 in August to 143 in January. Maximal daily temperature correlated negatively with the number of decompensations on the subsequent day (beta = −0.07 [95% confidence interval, −0.09 to −0.05], *p* < 0.001). The association remained significant in a multivariate linear regression model adjusted for other meteorological parameters (adjusted beta = −0.07 [−0.10 to −0.04], *p* < 0.001). Moreover, it was present across HF with preserved (n = 375; beta = −0.08 [−0.14 to −0.03], *p* = 0.004) and reduced (n = 331; beta = −0.08 [−0.13 to −0.02], *p* = 0.005) ejection fraction, but not with mildly reduced ejection fraction (n = 160; beta = −0.03 [−0.07 to 0.01], *p* = 0.200). **Conclusions:** In a European continental climate zone region, lower temperature was associated with a linear increase in subsequent HF decompensations. The sequelae of climate change on HF decompensations may burden healthcare systems in the future and should be systematically investigated in further studies.

## 1. Introduction

Heart failure (HF) remains a challenging clinical condition that strains the capacity of healthcare facilities all over the world. The prevalence of HF is up to 1–2% in adults and increases with age [1]. HF causes a significant amount in economic expenditure due to hospitalizations and medical treatment. Data from the United Kingdom estimates that the cost per year spent for the management of HF are about GBP 980 million, whereas the worldwide economic expenses per year treating HF are estimated at USD 108 billion, according to the World Bank [2]. Cardiac decompensations independently predict HF disease progression and prognosis in HF with preserved (HFpEF), mildly reduced (HFmrEF) and reduced ejection fraction (HFrEF) [3,4].

While HFpEF and HFrEF differ significantly with regard to demographics and etiology, cardiac decompensation in both entities is finally a consequence of fluid accumulation and intravascular congestion [5]. Several extrinsic factors have been identified in European guidelines as the most important triggers of cardiac decompensation, but weather influences have not been included here, probably due to a lack of sufficient evidence [6]. In fact, there are some suggestions that the risk of HF decompensations is affected by temperature. Cold temperature impairs exercise capacity, increases cardiac oxygen demand, and increases venous pressure [7]. Several studies in countries with other than continental climate have demonstrated that seasonal changes in weather, especially during cold periods, are associated with higher numbers of HF hospitalizations and increased mortality [5,8,9,10,11]. For instance, in a multicenter study conducted in Spain, a European country with Atlantic climate, a U-shaped association between temperature and risk of HF hospitalization on the subsequent day has been reported [5]. Furthermore, hospital admissions peak with relative temperature changes, e.g., when the temperature drops to its lowest point. This phenomenon can be observed in different climate zones regardless of absolute temperature values [12,13]. Yet, it remains unknown whether there is an impact of temperature on the risk of HF decompensations in countries with continental climate. Moreover, potential differences in temperature impact on heart failure subtypes have not been investigated yet.

In this study, we aimed to evaluate the association between daily temperature and the frequency of HF decompensations on the subsequent day in an emergency department (ED) in Eastern Austria, a Central European country with continental climate. Associations were also assessed within HF subtypes.

## 2. Materials and Methods

### 2.1. Study Design and Patient Population

This study was conducted as a systematic retrospective single center medical records review of admissions to the internal ED of an academic tertiary care center (University Hospital Graz, Graz, Austria) between August 2018 and July 2019. This hospital provides healthcare for 400,000 residents of the city, and is the maximum care provider for approximately 1.3 million people from surrounding areas.

The study was approved by the local Ethics Committee (EC-No. 32-087 ex 19/20) at the Medical University of Graz, and conducted in accordance with the Declaration of Helsinki for Good Clinical Practice. Informed consent was waived due to the retrospective character of the study. Patient data were pseudonymized in the dataset. A password-protected digital logbook connecting patient names with their unique identifiers was kept strictly confidential.

At this center, patient work-up is based on the Manchester Triage System, and comprises systematic assessment of lead symptoms, vital parameters, basic laboratory parameters, 12-lead electrocardiogram, documentation of prescribed medicines, and physical examination. Medical records of all patients presenting in the ED were assessed for the presence of decompensated HF, as reported previously [14]. Briefly, admissions were systematically screened for signs of heart failure and the presence of exertional or resting dyspnea. As recommended, heart failure was defined as the presence of at least one sign or presence of dyspnea and elevated levels of N-terminal pro-brain B-type natriuretic peptide (NT-proBNP) [15]. Case records with inconclusive medical reports were individually re-assessed by a heart failure specialist (NV) [1].

Transthoracic echocardiography was conducted at the discretion of the treating physician in the ED. Among the total HF cohort, left ventricular ejection fraction (LVEF) derived from transthoracic echocardiography was available in 866 admissions (69%), allowing for stratification into HFpEF, HFmrEF and HFrEF (Appendix A) [1]. Maximal daily temperature data and further meteorological data (humidity, daily precipitation, average wind speed, sunshine duration, mean atmospheric pressure) were obtained from the Austrian Central Institute for Meteorology and Geodynamics (ZAMG) and were derived from the meteorological station at the University of Graz located in the city center.

### 2.2. Statistical Methods

The analysis of temperature dependence on HF decompensation was performed using MATLAB Version 23. To assess the association between temperature and HF decompensations, a day-based approach was chosen (the day before the ED visit), based on the work by Miro and colleagues [8]. Maximal daily temperature (Tmax) was correlated with the respective daily total HF events and HF subtypes HFpEF, HFmrEF, and HFrEF using linear regression. In addition, HF decompensations per month were plotted against the average monthly temperature to assess the overall trend of these events. Maximal daily temperature was correlated with levels of high-sensitive C-reactive protein (hsCRP, 10-log transformed) using Pearson correlation.

Statistics of clinical baseline and meteorological data were performed with IBM SPSS Statistics Version 29 and R version 4.4.2. Continuous variables were reported as medians (25–75th percentile), while count data were listed as absolute frequencies (%). The significance level α was set at 5%.

## 3. Results

In the time between August 2018 and July 2019, 32,028 visits to the ED were registered overall. The cohort of the present study consisted of 1248 admissions presenting with signs and symptoms of HF decompensation, and were therefore eligible for further analysis. The median age was 80 (25–75th percentile: 74, 87) and 625 (50%) were female, as listed in Table 1. Echocardiographic examination was performed in 866 cases. Patients were stratified by their LVEF, into 375 (43%) patients with HFpEF, 160 (19%) with HFmrEF, and 331 (38%) with HFrEF, respectively. Median NT-proBNP levels were 4080 (1741, 9410 pg/mL, estimated glomerular filtration rate (eGFR) levels were 48 (31, 62) mL/min/1.73 m^2^ and CRP levels were 13 (5, 41) mg/L. While 47% of all admissions showed dyspnea, most present signs of cardiac decompensation were leg edema (62%), pleural effusion (59%), and pulmonary congestion (56%).

January and December were the coldest months with median temperatures of 3.9 (1.7, 6.2) °C and 6.3 (2.3, 8.4) °C, respectively. June and August were the overall hottest months, with median temperatures of 28.4 (27.8, 30.4) °C and 29.4 (26.7, 30.7) °C, respectively. Maximal daily temperature was stratified by deciles. In the lowest temperature decile (n = 33), temperature ranged from −1.5 °C to 4.4 °C, and it was composed of 16 days in January, 14 days in December, and 3 days in February. In the highest temperature decile (n = 36), temperature ranged from 30.3 °C to 37.2 °C, and it was composed of 14 days of August, 12 days of July, and 10 days of June.

The median frequency of HF decompensations per day was 3 (2, 5) during twelve months. Decompensations followed a seasonal trend, with a peak during the coldest months of December (n = 138) and January (n = 143), and a nadir during the hottest months of June (n = 65) and August (n = 65). The seasonal trend of temperature and HF decompensations is illustrated in Figure 1.

Maximal daily temperature correlated negatively with the number of decompensations on the subsequent day (beta = −0.07 [95% confidence interval, −0.09, −0.05], *p* < 0.001). The daily frequency of HF decompensations decreased by an average of 0.35 events per 5 °C temperature increase. The association remained significant in a multivariate model adjusted for day of the week, average humidity, daily precipitation, average wind speed, sunshine duration, and average atmospheric pressure (adjusted beta = −0.07 [−0.10, −0.04], *p* < 0.001; Table 2; Figure 2). The negative correlation between maximal daily temperature and subsequent HF decompensation was present in admissions with HFpEF (n = 375; beta = −0.08 [−0.14, −0.03], *p* = 0.004) and HFrEF (n = 331; beta = −0.08 [−0.13, −0.02], *p* = 0.005) ejection fraction, but not with HFmrEF (n = 160; beta = −0.03 [−0.07, 0.01], *p* = 0.200). Maximal daily temperature showed a weak negative correlation with hsCRP levels (Pearson r = −0.09, *p* < 0.01).

## 4. Discussion

This study is the first to show that lower daily temperature is associated with a higher frequency of subsequent HF decompensations in a continental climate zone region. This association was independent of other potentially confounding meteorological parameters, and was similar for patients with HFpEF and HFmrEF, suggesting that low temperature may predispose to cardiac congestion independently of the underlying HF subtype. Our study extends investigations from regions other than continental climate zone, underscoring that low temperature may adversely impact HF burden.

Our study results are overall comparable with previous works from other than continental climate regions in France, Spain, India, sub-Sahara and Japan. Boulay et al. published their results about the seasonal variation in chronic HF decompensations peaking in colder seasons in France. They demonstrated a significant increase of mortality in colder seasons [16]. Miro et al. recently published a study evaluating the influence of temperature and atmospheric pressure on the severity of HF decompensations in 16 Spanish cities, with predominantly Atlantic climate. Here, there was an increase in admissions, as well as severity of decompensations with temperatures above 20 degrees on the day before admission, for temperatures below 5.4 degrees only an increase in admissions, but not severity, was shown. They hypothesized that during cold seasons HF admissions increase often due to infections [8]. In climate zones with higher temperatures, there seems to be an impact of low temperatures on HF decompensations, as shown in 2021 by Nganou-Gnindjio in a sub-Saharan country with a borderline-to-significant correlation [10]. Yet, studies assessing the temperature influence on cardiac decompensations in a Central European Country with continental climate have been lacking. According to the Köppen classification, the city of Graz is representative of a continental climate zone, which is characterized by warm summers and coldest average temperatures below 0 °C. It is anticipated that in the context of climate change, extreme temperature peaks will increase in frequency [17,18]. Our observational results might be the foundation for future works to assess the influence of temperature drops on HF burden. In the context of climate change, our study may carry relevant implications and suggests, although causality cannot be inferred from these observational findings, that the sequelae of climate change on HF decompensations may have an impact on healthcare systems in the future.

The association between lower temperatures and higher frequency of HF decompensations persisted across the HF subgroups HFpEF, HFmrEF, and HFrEF. The lack of significance among the patients with HFmrEF is in our view mainly due to the low group size and resulting lack of power. Classically, HFpEF differs from HFmrEF and HFrEF by etiology, overall affecting older patients, more women and more patients with non-cardiovascular comorbidities [1]. Given the divergence in etiologies, pathophysiology and clinical appearance between HFpEF, HFmrEF, and HFrEF, it is interesting to see similar associations between temperature and HF decompensations regardless of HF subtype. Of note, the HF subtype was only available in 69% of patients, so a potential selection bias cannot be excluded and the results regarding HF subtypes should be interpreted reluctantly. Yet, the distribution of HF subtypes in our cohort equals other real-world data [19]. The relatively high hospitalization rate of 84% observed in this elderly cohort (median age 80 years) highlights the substantial medical resource requirements associated with cardiac decompensation, underscoring the clinical relevance of our findings.

Potential mechanisms relating low temperature with HF decompensations are not fully understood. Several environmental parameters besides temperature, such as humidity, wind speed or sunshine duration, show seasonal variation, but none was associated with HF decompensations indicating an independent impact of temperature. Importantly, air pollution increases in cold months and may also adversely affect the decompensation risk in HF patients. However, due to unavailability, measures of air pollution were not considered in our analyses [20]. The higher likelihood of HF decompensations during the weekend is most likely a consequence of closed private doctors’ offices, such as of general practitioners, during the weekend. Previous studies indicated that respiratory tract infections contribute significantly to HF decompensations and poor prognosis in patients with HF [15,21]. Indeed, one-fifth of all HF decompensations are a result of respiratory tract infections [15,16]. Yet, in our HF cohort, there was only a weak correlation between CRP levels and maximal daily temperature. This suggests that the contributing role of infections to the observed association between temperature and HF decompensations may be negligible in our cohort. Another potentially predisposing factor for HF decompensations with seasonal dependence may be acute coronary syndrome which was, however, not assessed in this study [22]. While there are studies suggesting that warm temperatures are favorable for hemodynamics in HF, little is known about cold temperature effects in chronic HF [23]. Exposure to cold temperatures is related with an increase in arterial pressure, which may in turn also impact HF burden [18]. Exposure to cold induces an increase in pulmonary artery pressure and systemic blood pressure both in hypertensive and non-hypertensive individuals [24,25]. Furthermore, cold exposure triggers activation of the sympathetic nervous system [26,27]. In fact, there is some evidence that hypertension-related pulmonary oedema could explain our observed association between cold temperatures and HF decompensations [28,29].

Our study carries some limitations. The observational design of the study precludes conclusions on causal relationships between temperature and HF hospitalizations. Echocardiography was only performed in 69% of admissions, potentially introducing a certain degree of selection bias. Due to the retrospective character of the study, comprehensive assessments of mechanisms underlying cardiac decompensation were beyond the scope of the study.

## 5. Conclusions

We observed an independent negative relationship between maximal daily temperature and the frequency of HF hospitalizations in an ED located in an Eastern Europe region with continental climate. Patients with HF should be educated to self-assess their congestion, for example, by regular weight measurements particularly during cold temperature periods. The sequelae of climate change on HF decompensations may impact healthcare systems in the future and should be systematically investigated in further studies.

## Figures and Tables

**Figure 1 biomedicines-13-01054-f001:**
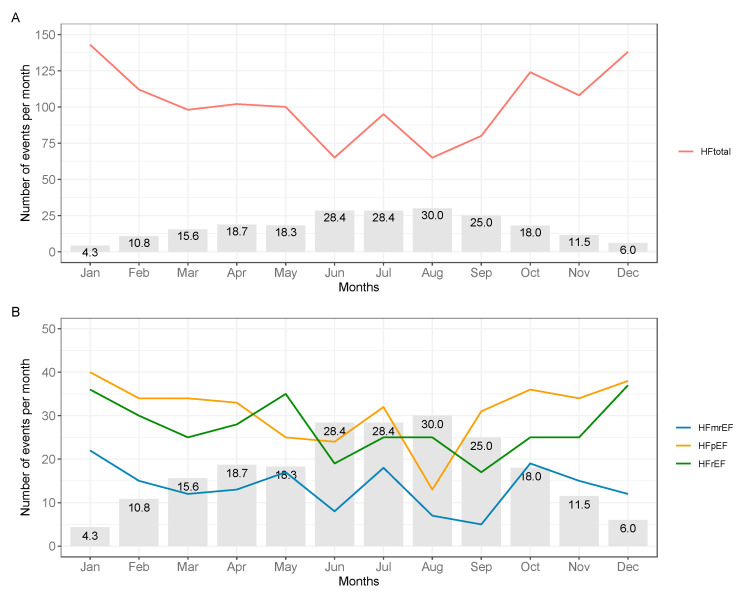
Frequency of heart failure decompensations per month across the total sample of heart failure decompensations (n = 1248, **A**), and heart failure subtypes (n = 866, **B**). Median maximal daily temperature is indicated for every month. Abbreviations: HFpEF, heart failure with preserved ejection fraction; HFmrEF, heart failure with mildly reduced ejection fraction; HFrEF, heart failure with reduced ejection fraction; HFtotal, total sample of heart failure decompensations.

**Figure 2 biomedicines-13-01054-f002:**
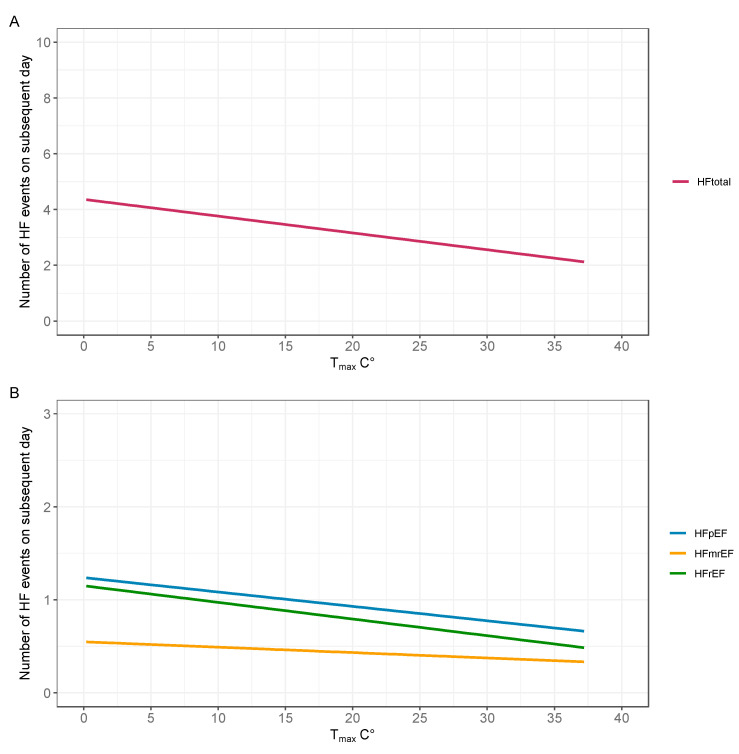
Association between maximal daily temperature and the number of heart failure decompensations on the subsequent day using linear model. Associations are depicted for the total cohort (n = 1248, **A**), and for cases with available echocardiography (n = 866, **B**) allowing for stratification into heart failure subtypes. Abbreviations: HFpEF, heart failure with preserved ejection fraction; HFmrEF, heart failure with mildly reduced ejection fraction; HFrEF, heart failure with reduced ejection fraction.

**Table 1 biomedicines-13-01054-t001:** Baseline characteristics of the total sample (n = 1248).

	n (% Miss)	Summary Estimate
Age—yrs.	1248 (0)	80 (74–87)
Females—n (%)	1248 (0)	625 (50)
Clinical features of heart failure		
HFpEF (EF ≥ 50%)—n (%)	866 (31)	375 (43)
HFmrEF (EF 41–49%)—n (%)	866 (31)	160 (19)
HFrEF (EF ≤ 40%)—n (%)	866 (31)	331 (38)
Hospitalization on admission—n (%)	1248 (0)	1045 (84)
Intravenous diuretic treatment on admission—n (%)	1246 (<1)	578 (46)
Heart failure signs and symptoms, n (%)		
Leg edema	1248 (0)	779 (62)
Pleural effusion	1248 (0)	738 (59)
Pulmonary congestion	1248 (0)	697 (56)
Pulmonary rales	1248 (0)	510 (41)
Dyspnea	1248 (0)	585 (47)
Other signs	1248 (0)	180 (14)
Laboratory parameters		
NT-proBNP—pg/ml	1137 (9)	4080 (1741–9410)
eGFR—mL/min/1.73 m^2^	1245 (<1)	48 (31–62)
hsCRP—mg/L	1096 (12)	13 (5–41)

Variables are median [25th–75th percentile] or n (%). Abbreviations: hsCRP = high sensitive C-reactive protein; eGFR = estimated glomerular filtration rate; EF = ejection fraction; HFmrEF = heart failure with mildly reduced ejection fraction; HFpEF = heart failure with preserved ejection fraction; HFrEF = heart failure with reduced ejection fraction; NT-proBNP = N-terminal pro-B-type natriuretic peptide.

**Table 2 biomedicines-13-01054-t002:** Univariate and multiple linear regression analysis of maximal daily temperature and other meteorological parameters and total heart failure decompensations on the subsequent day.

	Univariate Linear Regression			Multiple Linear Regression		
	Beta-Coefficient	95% CI	*p*-Value	Adjusted Beta-Coefficient	95% CI	*p*-Value
Maximal daily temperature	−0.07	−0.09, −0.05	<0.001	−0.07	−0.10, −0.04	<0.001
Day						
Monday	Reference			Reference		
Tuesday	−1.40	−2.2, −0.61	<0.001	−1.30	−2.1, −0.56	<0.001
Wednesday	−0.95	−1.8, −0.14	0.022	−0.88	−1.7, −0.10	0.028
Thursday	−0.77	−1.6, 0.04	0.064	−0.75	−1.5, 0.04	0.062
Friday	−0.58	−1.4, 0.24	0.2	−0.51	−1.3, 0.27	0.2
Saturday	−1.80	−2.6, −1.0	<0.001	−1.80	−2.6, −1.0	<0.001
Sunday	−1.10	−1.9, −0.26	0.010	−1.10	−1.9, −0.30	0.007
Average humidity	0.02	0.00, 0.03	0.060	−0.01	−0.03, 0.01	0.400
Daily precipitation	−0.02	−0.07, 0.03	0.500	0.01	−0.05, 0.07	0.700
Average wind speed	−0.10	−0.18, −0.02	0.013	−0.06	−0.15, 0.03	0.200
Daily sunshine duration	−0.09	−0.14, −0.04	<0.001	0.01	−0.07, 0.09	0.900
Average atmospheric pressure	−0.02	−0.05, 0.01	0.300	−0.02	−0.05, 0.01	0.200

Abbreviations: CI, confidence interval.

## Data Availability

The data presented in this study are available on request from the corresponding author.

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
