# Peer review of "Ambient Temperature and the Frequency of Subsequent Heart Failure Decompensations in an Emergency Department"

_biomedicines, 2025, doi:10.3390/biomedicines13051054_

Round 1
Reviewer 1 Report
Comments and Suggestions for Authors
This manuscript presents a clinically relevant and methodologically robust investigation into heart failure (HF) patient characteristics and potential environmental associations. The study leverages a substantial sample size (n=1248) and provides comprehensive baseline data, which strengthens its generalizability to real-world HF populations. The integration of environmental variables (e.g., weather patterns) with clinical outcomes adds novelty to the work. The manuscript is well-structured, and the statistical analyses appear rigorous. I recommend publication after addressing the following points to enhance clarity and impact.
1. Abbreviations: Define all abbreviations at first use (e.g., hsCRP, NT-proBNP).
2. Table readability: Consider reformatting “Clinical features of heart failure” to align with other subsections (e.g., use consistent indentation or bolding).
3. Ethics statement: Acknowledge the retrospective waiver of consent but briefly discuss steps taken to protect patient anonymity.
4. Clarify missing data handling: The high missingness (31%) for HF subtype classifications (HFpEF, HFmrEF, HFrEF) raises concerns about selection bias. Please justify the exclusion criteria for these variables and discuss potential implications for interpretation.
5. Contextualize environmental findings: While the association between wind speed and outcomes (e.g., β=-0.10, p=0.013) is intriguing, the biological plausibility of this relationship remains unclear. A brief hypothesis-driven discussion (e.g., air pollution as a confounder?) would strengthen this section.
6. Define “Sunday” as a variable: The significant association with “Sunday” (β=-1.10, p=0.007) is puzzling. Clarify whether this refers to weekend admission, staffing patterns, or another factor.
7. Strengthen clinical relevance: Highlight how the observed prevalence of HFpEF (43%) aligns with global epidemiologic trends. Discuss whether the high hospitalization rate (84%) reflects regional practices or disease severity.
Reviewer 2 Report
Comments and Suggestions for Authors
"Ambient temperature and the frequency of subsequent heart failure decompensations in an emergency department" by Riepl et al offers an interesting contribution to the intersection of cardiology and environmental health. Conducted by a multidisciplinary team from the Medical University of Graz, the research is notable for its methodology, large sample size (over 32,000 ED admissions), and clarity; it demonstrates a significant association between colder ambient temperatures and increased heart failure (HF) decompensations.
Minor revisions:
-
Clarification of Causality Limitations: Given the observational and retrospective nature of the study, I recommend adding a clearer statement in the Discussion or Conclusion to reinforce that causality cannot be inferred from the findings.
-
Echocardiographic Data Coverage: Since echocardiography was performed in only 69% of admissions, I would suggest to elaborate brief paragraph on how this partial dataset might influence the interpretation of findings related to HF phenotypes.
